# Model Alignment Search

## Abstract

When can we say that two neural systems are the same? What nuances do we miss when we fail to causally probe the representations of the systems? In this work, we introduce a method for connecting neural representational similarity to behavior through causal interventions. The method learns transformations that find an aligned subspace in which behavioral information can be interchanged between multiple distributed networks' representations. We first show that the method can be used to transfer the behavior from one frozen Neural Network (NN) to another in a manner similar to model stitching, and we show how the method can differ from correlative similarity measures like Representational Similarity Analysis. Next, we empirically and theoretically show how the method can be equivalent to model stitching when desired, or it can take a form that has a more restrictive focus to shared causal information; in both forms, it reduces the number of required matrices for a comparison of n models to be linear in n. We then present a case study on number-related tasks showing that the method can be used to examine specific subtypes of causal information, and we present another case study showing that the method can reveal toxicity in fine-tuned DeepSeek-r1-Qwen-1.5B models. Lastly, we show how to augment the loss with a counterfactual latent auxiliary objective to improve causal relevance when one of the two networks is causally inaccessible (as is often the case in comparisons with biological networks). We use our results to encourage the use of causal methods in neural similarity analyses and to suggest future explorations of network similarity methodology for model misalignment.

## 1 Introduction

An important question for understanding both Artificial and Biological Neural Networks (ANNs and BNNs) is knowing what it means for one distributed system to model or represent another (Sucholutsky et al., 2023). Establishing isomorphisms between different distributed systems can be useful for simplifying their complexity and for understanding otherwise opaque inner mechanisms. Instead of asking how to interpret neural activity directly, we can instead use simplified neural systems, or neural systems understood through a higher level of analysis, and compare the internals of these known systems to uknown systems (Cao & Yamins, 2021; 2024; Richards et al., 2019). Furthermore, there are a number of open questions about how representations differ or converge between architectures, tasks, and modalities (Huh et al., 2024; Sucholutsky et al., 2023; Wang et al., 2024; Li et al., 2024; Hosseini et al., 2024; Zhang et al., 2024; Grant et al., 2025). Implicitly intertwined in these issues is the notion of network similarity, or dissimilarity. How do we measure the degree to which one model represents another? Researchers often use correlational methods to measure the similarity of different neural representations. We can see examples of this in works that perform direct correlational analyses between individual ANN activations and BNN firing rates (Yamins & DiCarlo, 2016; Maheswaranathan et al., 2019; Khosla & Williams, 2023; Williams et al., 2022), and in works that use Representational Similarity Analysis (RSA)—or Centered Kernel Alignment (CKA) (Kornblith et al., 2019; Williams, 2024)—finding 2nd order isomorphisms between model and system (Kriegeskorte et al., 2008). We also see examples of this in linear decoding techniques, where linear decodability and predictability can be used as a measure of the type of information encoded in distributed representations (Chen et al., 2020; Radford et al., 2021; Grill et al., 2020; Caron et al., 2021; Haxby et al., 2001; Haxby, 2013; Lampinen et al., 2024). A question remains, however, how to causally associate these neural analyses with behavioral outcomes. What do we miss when we ignore the behavioral relevance of the neural activity (Pearl, 2010; Geiger et al., 2024; Cloos et al., 2024;

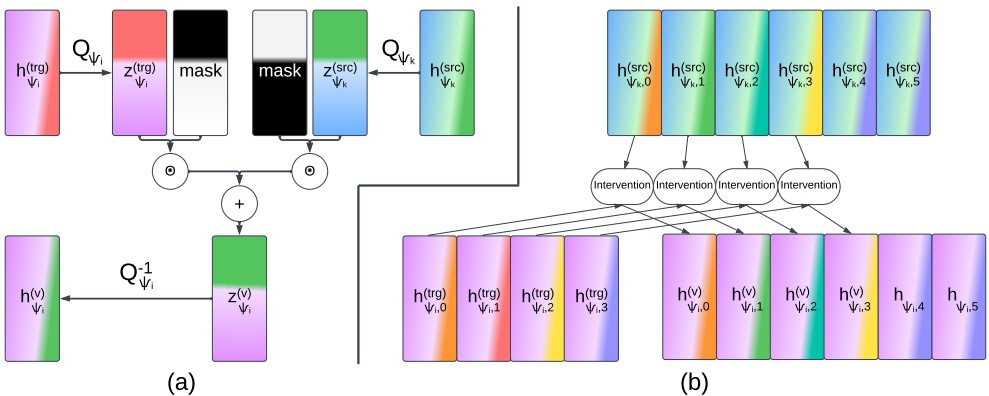

Figure 1: **(a)** A depiction of an interchange intervention from Equation 6 on the target latent vector $h_{\psi_i}^{(trg)}$ from model $\psi_i$ using the source latent vector $h_{\psi_k}^{(src)}$ from $\psi_k$. Rectangles represent vectors; colors distinguish between behaviorally relevant and extraneous activity. The causally relevant information is spread across the neural population (Smolensky, 1988; Park et al., 2023), represented by the red and green semi-vertical slices in the respective $h$ vectors. To disentangle and isolate the complement of the behavioral null-space, we rotate the $h$ vectors into an aligned space, using $Q_{\psi_i}$ and $Q_{\psi_k}$, where the behaviorally relevant information is organized along separate dimensions than the extraneous, behavioral null-space. We can then intervene on and transfer the relevant information without affecting other information. In the figure, this is done by applying binary masks (black represents 0s and white 1s and $\odot$ is a Hadamard product) to the $z$ vectors and taking their sum. We then invert the rotation on $z_{\psi_i}^{(v)}$ to return it to the original latent space where it can be used by $\psi_i$ to make predictions. **(b)** Depicts Stepwise MAS where the individual intervention shown in panel (a) is applied at multiple token positions in the sequence. We limit our interventions to contiguous sets of tokens starting from the first token and ending with a sampled position $t$.

Schaeffer et al., 2024)? And how do we incorporate a focus on behavioral relevance in our measures of neural similarity?

Some works have made progress towards connecting representational similarity to behavior by transforming intermediate representations from one neural system into a usable form for another. We see examples in works like Sexton & Love (2022) where they attempt to use transformed neural recordings in a trained computational model, and in *model stitching*, where a linear mapping is learned from intermediate representations from one ANN to another for the purpose of measuring similarity and/or improving one of the two models (Lähner & Moeller, 2024; Moschella et al., 2023; Bansal et al., 2021; Lenc & Vedaldi, 2015). To build upon these works, we ask: 1) what do these representational mappings tell us about the underlying representations of the two systems; are behaviorally successful mappings both necessary and sufficient for claims of functional similarity? 2) How can we isolate and compare the behaviorally relevant activity within transferred representations? 3) How do we isolate the similarity of specific behavioral information in the representations, thus allowing us to compare models with disparate behavioral outputs? And 4) how do we achieve causal relevance in systems for which we do not have causal access—as is often the case in ANN to BNN comparisons?

In this work, we introduce a neural similarity method, *Model Alignment Search* (MAS), that uses causal interventions to isolate and compare behaviorally relevant activity from neural representations in different ANNs. MAS learns a rotation matrix, or more generally an alignment function, for each model in the analysis to simultaneously uncover causally relevant latent subspaces in each model and map these spaces onto one another so that behavioral information can be patched between models. These manipulated representations are then returned back to their original neural space where the networks' resulting behavior can be compared to the expected counterfactual behavior as a means of measuring neuro-behavioral similarity.

The contributions of this work are as follows.

1. We introduce and motivate MAS by comparing to correlative representational similarity methods demonstrating a need to supplement correlational with causal methods depending on the research goals.

2. We empirically and theoretically compare MAS to model stitching, showing that it requires less compute when comparing 3 or more models, and it can be more restrictive to causal subspaces than model stitching.

3. Using numeric cognition as a case study, we show how MAS can be used to examine the similarity of specific types of behavioral information in neural representations.

4. Using model misalignment as a case study, we show that the method can be used to reveal toxicity in DeepSeek-R1-Qwen-1.5B models (Guo et al., 2025) finetuned to be toxic or nontoxic.

5. We introduce a *counterfactual latent* auxiliary loss objective that can be used to find behaviorally relevant alignments in cases where one of the two models is causally inaccessible (making the technique potentially relevant for comparisons between ANNs and BNNs).

## 2 Background and Related Work

There has been a call to use causal methods for both interpreting neural activity and making comparisons between different NN internals (Geiger et al., 2024; Sexton & Love, 2022; Feather et al., 2025; Lampinen et al., 2025). This call has been made due to principles of causal mediation Pearl (2010), and, more recently, representation and causal computation have been shown to be different in NNs (Hermann & Lampinen, 2020; Braun et al., 2025; Lampinen et al., 2025). In the mechanistic interpretability literature, a popular method for making claims about cause and effect in NNs is activation patching (Geiger et al., 2020; Vig et al., 2020; Wang et al., 2022; Meng et al., 2023) where experimentalists manipulate the NN activity to observe downstream effects. One such method is DAS, which finds causal subspaces that align with high-level variables from causal abstractions (Geiger et al., 2021; 2023; Wu et al., 2024). In cases of multi-model comparisons, a particularly popular method for causally comparing representations between different models is model stitching (Lenc & Vedaldi, 2015; Bansal et al., 2021). In this section, we present a background on both model stitching and DAS to establish notation and a conceptual framework for MAS.

### 2.1 Model Stitching

Model stitching aims to learn a mapping from the latent state space of one NN, $\psi_1$, to that of another, $\psi_2$. Let $h_{\psi_i} \in \mathbb{R}^{d_{\psi_i}}$ denote a latent representation from $\psi_i$; $i \in \{1, 2\}$. Stitching seeks a transformation $T : \mathbb{R}^{d_{\psi_1}} \to \mathbb{R}^{d_{\psi_2}}$ such that

$$T(h_{\psi_1}) \approx h_{\psi_2}$$

where $h_{\psi_2} \in \mathbb{R}^{d_{\psi_2}}$ is the corresponding representation in $\psi_2$. The mapped representation $T(h_{\psi_1})$ is then fed into $\psi_2$'s remaining layers, denoted $f_{\psi_2}(x, h)$, so that $\psi_2$ can make predictions consistent with those produced by $\psi_1$.

Although $T$ can be trained in a number of ways, we focus on cases in which $T$ is learned by optimizing $\psi_2$'s outputs, conditioned on the stitched representation $\hat{h}_{\psi_2} = T(h_{\psi_1})$, to match $\psi_1$'s behavior, also conditioned on $h_{\psi_1}$. The learning objective is a behavioral matching loss:

$$\mathcal{L}(T) \;=\; \mathbb{E}_{x \sim \mathcal{D}}\big[\ell\big(f_{\psi_1}(x, h_{\psi_1}) \,\|\, f_{\psi_2}(x, \hat{h}_{\psi_2})\big)\big] \tag{1}$$

where $x$ $\mathcal{D}$ are samples from a dataset and $\ell$ is a divergence measure (e.g., KL divergence or cross-entropy) between $\psi_1$'s output distribution and $\psi_2$'s output distribution under the stitched representation.

In this work, we narrow our focus to transformations of the form

$$T(h_{\psi_1}) = aW h_{\psi_1}$$

where $a \in \mathbb{R}$ is a scalar constant and $W \in \mathbb{R}^{d_{\psi_2} \times d_{\psi_1}}$ is a matrix—possibly low rank—with orthonormal rows.

Model stitching has been used as a method to make claims about the similarity between two different NNs Bansal et al. (2021). However, it has problems as a similarity method due to the fact that it is unidirectional, and, as we will show in Section 4.2, it can make use of $\psi_1's$ behavioral null space when mapping to $\psi_2$—where the behavioral null space is defined as the span of vector directions that have no influence on the NN's behavior. Furthermore, a successful mapping ignores naturally occurring variability in $\psi_2$'s behavioral null space. Thus, a successful mapping $T$ only needs to produce a single sufficient representation for any given behavior. Thus, successful $T$'s do not necessarily indicate that $\psi_1$ and $\psi_2$ perform the task in the same way.

## 2.2   DAS Formulation

DAS is a framework for causally determining the degree of alignment between a single NN's latent representations and variables from a causal abstraction (Geiger et al., 2021; 2023; Wu et al., 2024). We can think of such variables as those in a computer program, or nodes in a Directed Acyclic Graph (DAG). DAS does this alignment by attempting to transform model latent states $h \in \mathbb{R}^{d_\psi}$ into aligned vectors $z \in \mathbb{R}^{d_\psi}$ that consist of contiguous subspaces that have an analogous causal effect on behavior as high-level variables from a pre-specified causal abstraction. The transformation is performed by learnable, invertible *Alignment Function* (AF), $\mathcal{A}$, as follows: $z = \mathcal{A}(h)$ (Grant et al., 2025). We narrow our focus to orthogonal AFs: $\mathcal{A}(h) = Qh$ where $Q \in \mathbb{R}^{d_\psi \times d_\psi}$ is an orthogonal matrix. The purpose of this transformation is that it organizes causal subspaces into separated, contiguous dimensions in $z$. This allows for causal interventions on the values of specific subspaces—analogously to changing value of a variable in a computer program—without affecting the values of the other subspaces.

Concretely, for a given causal abstraction with variables $\text{var}_i \in \{\text{var}_1, \text{var}_2, ..., \text{var}_n\}$, DAS learns a rotation $Q$ to find a $z$ composed of subspaces $\vec{z}_{\text{var}_i} \in \mathbb{R}^{d_{\text{var}_i}}$ corresponding to each of the variables of the causal abstraction. We also include a causally irrelevant subspace, $\vec{z}_{\text{extra}} \in \mathbb{R}^{d_{\text{extra}}}$, encoding extraneous, behaviorally irrelevant activity (i.e., the behavioral null-space and dormant subspaces—defined as those that do not vary between inputs (Makelov et al., 2023)).

$$Q(h) = z = \begin{bmatrix} \vec{z}_{\text{var}_1} \\ \vec{z}_{\text{var}_2} \\ \cdots \\ \vec{z}_{\text{var}_n} \\ \vec{z}_{\text{extra}} \end{bmatrix} \qquad (2)$$

Each $\vec{z}_{\text{var}_i} \in \mathbb{R}^{d_{\text{var}_i}}$ is a column vector of potentially different lengths satisfying the relation $d_{\text{extra}} + \sum_{i=1}^{n} d_{\text{var}_i} = d_\psi$. Under this assumption, the value of a single causal variable encoded in $h$ can be freely exchanged by performing an interchange intervention defined as follows:

$$h^{(v)} = Q^{-1}((1 - D_{\text{var}_i})Qh^{(trg)} + D_{\text{var}_i}Qh^{(src)}) \qquad (3)$$

Where $D_{\text{var}_i} \in \mathbb{R}^{d_\psi \times d_\psi}$ is a manually chosen, diagonal, binary matrix with $d_{\text{var}_i}$ contiguous ones along the diagonal used to isolate the dimensions that make up $\vec{z}_{\text{var}_i}$, $h^{(src)}$ is the *source vector* from which the subspace activity is harvested, $h^{(trg)}$ is the *target vector* into which the harvested activations are substituted, and $h^{(v)}$ is the resulting intervened vector that is used to replace $h^{(trg)}$ in the model's processing. This allows the model to make predictions using a different value of subspace $\text{var}_i$ after a successful intervention.

DAS uses *counterfactual behavior* to create intervention data to train and evaluate $Q$. Counterfactual behavior is defined as the behavior that *would* have occurred had the value of a variable in a causal abstraction been different and everything else remained the same (Pearl, 2010; Geiger et al., 2021). We can create intervention data by freezing the state of an environment at a particular point in time, changing one or more values in the causal abstraction's variables, and then unfreezing the environment and using the causal abstraction to generate the counterfactual behavior. These intervention samples can then be used as training labels to train $Q$ while keeping the NN parameters frozen. Once $Q$'s training has converged, the robustness of the alignment can be evaluated using the Interchange Intervention Accuracy (IIA)—defined as the NN's prediction accuracy on the counterfactual behavior for a held out set of interventions. IIA is then used to make claims about the NN's internal mechanisms.

# 3 Methods

All of the analyses in this work are performed on autoregressive, sequence based ANNs trained to predict sequences of tokens. Depending on the experiment, we consider GRUs (Cho et al., 2014), LSTMs (Hochreiter & Schmidhuber, 1997), shallow transformers (Vaswani et al., 2017; Su et al., 2023), and DeepSeek-R1-Distill-Qwen-1.5B (Guo et al., 2025). The majority of our analyses are confined to models trained to perform variations of numeric tasks. These tasks serve as a simplified setting to demonstrate how MAS works and how it compares to other methods. We additionally include a more general language modeling experiment where we explore the similarity of toxic vs nontoxic models as a case study on how to use MAS in more practical settings (Appendix B.1).

## 3.1 Numeric Tasks

Each task consists of a sequence of tokens that start with a beginning of sequence token, B, and end with an end of sequence token, E. Some of the tokens in each task are produced by the task environment and define the specific goals within the task. Other tokens are determined by these task provided tokens. Each trial is considered correct when all deterministic tokens are correctly predicted. During the model training, we include all token types in a Next-Token Prediction (NTP) cross entropy loss, even though many tokens are unpredictable. See Figure 3(c) for a visual depiction of each of the tasks described in this section.

**Multi-Object Task:** The environment presents some number of demonstration (demo) tokens that are each sampled with replacement from the set $\{D_a, D_b, D_c\}$. The task is to produce the same number of response (R) tokens as D tokens and end with an E token. The environment signals the end of the D tokens by producing a trigger (T) token. The number of D tokens at this point is referred to as the *object quantity* for the trial, which is uniformly sampled from 1 to 20 at the beginning. The set of possible tokens includes $\{B, D_a, D_b, D_c, T, R, E\}$. An example sequence with an object quantity of 2 is: "B $D_c$ $D_a$ T R R E"

To prevent solutions that use positional readout (Grant et al., 2025), we modify the task for transformer trainings: each token in the demo phase has a 0.2 probability of being a "void" token type, V, that has no impact on the object quantity. An example sequence with an object count of 2 could be: "B V $D_b$ V V $D_c$ T R R E". All evaluations and analyses use the original Multi-Object Task.

**Same-Object Task:** same structure as the Multi-Object task except that all D and R tokens are replaced by a single token type, C. The set of possible tokens includes $\{B, C, T, E\}$. An example sequence with an object quantity of 2 would be: "B C C T C C E".

**Modulo:** This task is similar to the Multi-Object task except the number of R tokens is equal to the object quantity mod 4. An example trial could be, "B $D_b$ $D_c$ $D_a$ $D_c$ $D_b$ T R E".

**Rounding:** Similar to the Modulo task except the number of R tokens is equal to the object quantity rounded to the nearest multiple of 3. An example trial could be, "B $D_b$ $D_c$ $D_a$ $D_c$ T R R R E".

## 3.2 Model Architectures

In our numeric task experiments, each model, $\psi$, is autoregressively trained to perform only one of the tasks through next-token prediction (NTP). We train 2 model seeds for each task variant up to $> 99.99\%$ accuracy on both training and validation data and freeze the weights before analysis and interpretation. We consider Gated Recurrent Units (GRUs) (Cho et al., 2014), Long-Short Term Memory recurrent networks (LSTMs) (Hochreiter & Schmidhuber, 1997), and two layer Transformers based on the Roformer architecture (Vaswani et al., 2017; Touvron et al., 2023; Su et al., 2023). The custom GRUs and Transformers use a dimensionality of 128, whereas the LSTM uses 64 dimensions for each the h and c vectors. We leave details of GRU and LSTM cells to the referenced papers beyond noting that the GRU and LSTM based models have the structure:

$$h_{t+1} = g(h_t, x_t) \tag{4}$$
$$\hat{x}_{t+1} = f(h_{t+1}) \tag{5}$$

Where $h_t$ is the hidden state vector at step $t$, $x_t$ is the input token at step $t$, $g$ is the recurrent function (either a GRU or LSTM cell), and $f$ is a two layer (two matrix) feed-forward network (FFN) used to make a prediction, $\hat{x}_{t+1}$, of the token at step $t+1$ from the updated hidden state $h_{t+1}$.

The transformer architecture uses Rotary Positional Encodings (RoPE) (Su et al., 2023) and GELU nonlinearities (Hendrycks & Gimpel, 2023). Transformers use a history of input tokens, $X_t = [x_1, x_2, ..., x_t]$, at each time step, $t$, to make a prediction: $\hat{h}_t = g(X_t)$, and $\hat{x}_{t+1} = f(h_t)$ where $g$ and $f$ are now sets of transformer layers. We show results from 2 layer, single attention head transformers. We refer readers to Figure 4 and Appendix B.2 for more details. For our experiments on DeepSeek-R1-Distill-Qwen-1.5B, we refer readers to Guo et al. (2025); Yang et al. (2025) for architectural details and Appendix B.1 for finetuning details.

### 3.3 MAS Formulation

At a high level, MAS can be thought of as a combination of model stitching and DAS. MAS causally measures the degree to which multiple models' behavioral subspaces align with each other.

Using our notation from Sections 2.1 and 2.2, MAS learns an invertible alignment function, $\mathcal{A}_{\psi_i} : \mathbb{R}^{d_{\psi_i}} \to \mathbb{R}^{d_{\psi_i}}$ for each $\psi_i$ of $\mathbb{N}$ models in the analysis, such that $\mathcal{A}_{\psi_i}(h_{\psi_i}) = z_{\psi_i}$, and within each $z_{\psi_i}$, MAS performs interchange interventions to examine the degree to which the subspaces are causally interchangeable for all included models. It does this using both between and within model interchange interventions. Effectively, MAS asks, does $\vec{z}_{\psi_i, \text{var}_k} = \vec{z}_{\psi_j, \text{var}_k} \forall (i \in \mathbb{N}, j \in \mathbb{N})$? We limit our empirical analyses to $\mathbb{N} = 2$ and we only examine cases where $\mathcal{A}_{\psi_i} = Q_{\psi_i} = a_{\psi_i} U_{\psi_i}$ where $a_{\psi_i}$ is a scalar and $U_{\psi_i} \in R^{d_{\psi_i} \times d_{\psi_i}}$ is an orthogonal matrix.

With these intutions, we can generalize Equation 3 to the multi-model case for an intervention on the subspace for $\text{var}_k$:

$$h_{\psi_i}^{(v)} = Q_{\psi_i}^{-1}((1 - D_{\psi_i, \text{var}_k})Q_{\psi_i}h_{\psi_i}^{(trg)} + D_{\psi_j, \text{var}_k}Q_{\psi_j}h_{\psi_j}^{(src)}) \tag{6}$$

Where $i$ and $j$ can be any model index in the set of all models considered, $Q_{\psi_i}$ is a scaled orthogonal rotation matrix for $\psi_i$, $D_{\psi_i, \text{var}_k} \in R^{d_{\psi_i} \times d_{\psi_i}}$ is a diagonal, binary matrix with $d_{\text{var}_k}$ non-zero elements used to isolate the dimensions corresponding to $\vec{z}_{\text{var}_k}$, $h_{\psi_j}^{(src)}$ is the *source vector* from which the subspace is harvested, $h_{\psi_i}^{(trg)}$ is the *target vector* into which activity is substituted, and $h_{\psi_i}^{(v)}$ is the resulting intervened vector that replaces $h_{\psi_i}^{(trg)}$ in $\psi_i^{(trg)}$'s processing, allowing the model to make causally intervened predictions. See Figure 1(a) for a visualization.

For many of our analyses, we will perform MAS without segregating the behaviorally relevant subspace. Concretely, this can be formalized as a causal abstraction that uses a single Full variable in which all behaviorally relevant information is encoded, $\vec{z}_{full}$, and all extraneous information is encoded in $\vec{z}_{extra}$, the behavioral null-space and dormant subspaces. For completion: $z_{\psi_i} = \begin{bmatrix} \vec{z}_{\psi_i, full} \\ \vec{z}_{\psi_i, extra} \end{bmatrix}$, and we will freely isolate and manipulate $\vec{z}_{\psi_i, full}$ using Equation 3.

### 3.4 MAS Training

The MAS training procedure is similar to DAS in that the training loss is created using the output distribution from $\psi_i^{(trg)}$ conditioned on $h_{\psi_i}^{(v)}$ as predictions and the counterfactual behavior of the causal abstraction as labels. In the case of $\vec{z}_{full}$, the counterfactual behavior is the same as that of model stitching from Equation 1, where the training labels are the behavior from $\psi_{src}$ following the intervention. Using $y$ to denote the counterfactual behavior labels in the loss for a single intervention, the loss for a single intervention direction is as follows:

$$\mathcal{L}(Q_{\psi_i}^{(trg)}, Q_{\psi_j}^{(src)}) = \mathbb{E}_{x, y \sim \mathcal{D}}[\ell(f_{\psi_i}(h_{\psi_i}^{(v)}) \| y)] \tag{7}$$

In Sections 3.5 and 4.5 we will address cases in which we only include a subset of the intervention direction permutations in the training loss. However, unless otherwise stated, we default to averaging over all

permutations. The total loss for a single matrix $Q_{\psi_i}$ is as follows:

$$\mathcal{L}_{total}(Q_{\psi_i}) \;=\; \frac{1}{2\mathbb{N}-1} \sum_{j=1}^{\mathbb{N}} \big( \mathcal{L}(Q_{\psi_i}, Q_{\psi_j}) + \mathbf{1}_{\{i \neq j\}} \mathcal{L}(Q_{\psi_j}, Q_{\psi_i}) \big) \tag{8}$$

When using auto-differentiation frameworks (Paszke et al., 2019), we simply use the gradient from the mean of the losses in all directions:

$$\mathcal{L}_{total} \;=\; \frac{1}{\mathbb{N}^2} \sum_{i=1}^{\mathbb{N}} \sum_{j=1}^{\mathbb{N}} \mathcal{L}(Q_{\psi_i}, Q_{\psi_j}) \tag{9}$$

It is crucial to include interventions where $i = j$ as this adds a soft constraint to the alignment that encourages the separation of $\vec{z}_{full}$ from $\vec{z}_{extra}$.

**MAS Evaluation:** We can evaluate the quality of each $Q_{\psi_i}$ using the accuracy of the models' predictions on held out counterfactual data following the interventions. We always report the IIA of the worst performing causal direction in the analysis. In the numeric tasks experiments, a trial is considered correct when all deterministic tokens are predicted correctly using the argmax over logits. For the LLM toxicity experiments, we report token prediction accuracy rather than trial accuracy. We calculate error bars as Standard Error over unique model seed pairings. We include two additional seeds for within training type comparisons in the numeric task experiments, and three seeds for each of the DeepSeek finetunings.

### 3.5 Mas Variants

We explore a number of variations on the MAS procedure as a means of demonstrating how to use MAS in different situations for different purposes. We enumerate these variants in this section.

**Stepwise MAS**: Up to this point, we have described a form of MAS that performs interventions on latent vectors at individual time points in the sequence. We include an exploration of a variant in which Equation 6 is applied at each time point from the beginning of the sequence up to some time point $t$. For all embedding layer analyses, we use this variant. See Figure 1(b) for a visual depiction.

**Counterfactual Latent MAS (CLMAS)**: An interesting use case for neural similarity techniques is the comparison of ANNs with BNNs. An issue with causal comparisons of this nature is that we often have no causal access to the BNN in the analysis. We do, however, usually have neural recordings from the BNN. Using a tilde to denote causally inaccessible models, $\tilde{\psi}_i$, we can simulate such ANN-BNN comparisons by omitting causal interventions using $\tilde{\psi}_i$ as the target model from the training loss. We will assume, however, that we can still read from the latent vectors produced by $\tilde{\psi}_i$—analogous to a neural recording. We adapt Equation 8 for $\psi_k$, where $k \neq i$, to ignore the behavior of $\tilde{\psi}_i$ as follows:

$$\mathcal{L}_{accessible}(Q_{\psi_k}) = \frac{1}{2\mathbb{N}-1} \big( \mathcal{L}(Q_{\psi_k}^{(trg)}, Q_{\tilde{\psi}_i}^{(src)}) + \sum_{j=1}^{\mathbb{N}} \mathbf{1}_{\{j \neq i\}} \big( \mathcal{L}(Q_{\psi_k}^{(trg)}, Q_{\psi_j}^{(src)}) + \mathbf{1}_{\{j \neq k\}} \mathcal{L}(Q_{\psi_j}^{(trg)}, Q_{\psi_k}^{(src)}) \big) \big) \tag{10}$$

And the loss for $\tilde{\psi}_i$ is $\mathcal{L}_{inaccessible}(Q_{\tilde{\psi}_i}) = \frac{1}{2\mathbb{N}-1} \sum_{j=1}^{\mathbb{N}} \mathbf{1}_{\{j \neq i\}} \mathcal{L}(Q_{\psi_j}^{(trg)}, Q_{\tilde{\psi}_i}^{(src)})$.

We will show in Section 4.5 that if we attempt to train both $Q_{\tilde{\psi}_i}$ and $Q_{\psi_j} \forall j \in \mathbb{N}_{j \neq i}$ using only the behavioral gradient from minimizing Equation 10, the resulting $Q$'s do not achieve high IIA when evaluating causal interventions that use $\tilde{\psi}_i$ as the target model. To address this shortcoming, we introduce an auxiliary loss function to encourage causal relevance for $\tilde{\psi}_i^{(trg)}$. This auxiliary objective relies on what we will refer to as *Counterfactual Latent (CL) vectors*, which we define as latent vectors that encode the values of the causal variables that we would expect to exist in the intervened vector—$h_{\tilde{\psi}_i}^{(v)}$ from Equation 3—after an interchange intervention. If we focus on the Full variable case of $\vec{z}_{full}$, the CL vectors are prerecorded representations that produce to the same behavior as the counterfactual behavior training labels. We can obtain CL vectors by searching through a dataset of prerecorded $h_{\tilde{\psi}_i}$ vectors for cases that either have the correct variable values produced by the causal abstraction or those that lead to the same behavior as the counterfactual behavior.

As an example, if we have a causal abstraction with variables $var_1$, $var_2$, and $var_{extra}$, and, we expect $h^{(v)}_{\tilde{\psi}_i}$ to have a value of $y$ for variable $var_1$ and $w$ for variable $var_2$ after applying Equation 6, then a valid CL vector, $h^{(CL)}_{\tilde{\psi}_i}$, would be a pre-recorded representation in which the causal abstraction has labeled $h^{(CL)}_{\tilde{\psi}_i}$ to have the variable values: $var_1 = y$ and $var_2 = w$.

The auxiliary loss $\mathcal{X}$ for a single intervention sample is composed of an L2 loss and a cosine loss using CL vectors as the ground truth:

$$\mathcal{X}_{L2} \quad = \quad \frac{1}{2}||h^v_{\tilde{\psi}_i} - h^{(CL)}_{\tilde{\psi}_i}||^2_2 \tag{11}$$

$$\mathcal{X}_{cos} \quad = \quad -\frac{1}{2}\frac{h^v_{\tilde{\psi}_i} \cdot h^{(CL)}_{\tilde{\psi}_i}}{||h_{\tilde{\psi}_i}||_2\,||h^{(CL)}_{\tilde{\psi}_i}||_2} \tag{12}$$

where $h^{(v)}_{\tilde{\psi}_i}$ is the intervened target vector for the causally inaccessible model. The total CLMAS training loss is a weighted sum of the loss from Equation 10 and the auxiliary loss where $\epsilon$ is a hyperparameter: $\mathcal{L}_{CL} = \epsilon(\mathcal{X}_{L2} + \mathcal{X}_{cos}) + (1 - \epsilon)\mathcal{L}_{accessible}$. We show results from the best performing training of $\epsilon$ values 0.5, 0.89, 0.94, and the best $d_{full}$ out of 32, 64, and 128 dimensions.

**Transferring Specific Variables** Through our choice of counterfactual training behavior, we can explore alignment of specific subspaces between models that do not possess the same domain or co-domain. We focus on representations of number as a case study for their precise but general nature. To do this, we narrow the MAS interchange interventions to $\vec{z}_{\psi_1,numeric}$ and $\vec{z}_{\psi_2,numeric}$ by constructing counterfactual sequences for each $\psi$'s co-domain. We assume that the models encode a single numeric variable in the Multi-Object, Same-Object, Modulo, and Rounding tasks Grant et al. (2025). In the arithmetic task, we assume there is a numeric representation for the number of remaining operations, **Rem Ops**, and another representation for the cumulative value, **Cumu Val**. See expanded details in Appendix B.7.

**Model Stitching:** As baselines, we include causal intervention methods that we refer to as **Latent Stitch** and **Stitch**. Latent Stitch consists of a single orthogonal matrix $Q$ for two models, $\psi_1$ and $\psi_2$, that learns to map $h^{(src)}_{\psi_1}$ to $h^{(CL)}_{\psi_2}$ by minimizing the CL auxiliary loss from Equation 11 without including the behavioral objective Equation 10. Stitch consists of a single, possibly low rank, orthogonal matrix $Q$ trained in a single behavioral direction from $\tilde{\psi}_1$ to $\psi_2$ without using the CL auxiliary loss. Unless otherwise specified, assume $Q$ is full rank and the IIA is reported from validation intervention data in the $\tilde{\psi}_1$ to $\psi_2$ direction only.

### 3.6 Representational Similarity Analysis (RSA)

For a given model layer, we run the model on a batch of sequences consisting of 15 sequences from each object quantity 1-20. We then sample 1000 representational vectors uniformly from all time points excluding padding and end of sequence tokens. We construct a Representational Dissimilarity Matrix (RDM) as 1 minus the cosine similarity matrix over each pair-wise comparison of the representations (resulting in an RDM of dimensions $1000 \times 1000$). We create an RDM for two models and compare the RDMs using Spearman's rank correlation on the lower triangle of each matrix (Virtanen et al., 2020). We perform the RDM sampling 10 times and report the average over all 10 correlations. See Appendix B.5 for CKA methods.

## 4 Results

### 4.1 The Importance of Causal Analyses

We first set out to demonstrate why MAS could be preferable to a correlative similarity method such as RSA. RSA and CKA are second order correlational methods that examine the similarity between sample correlation matrices constructed from two models' representations (see Appendix B.4 for details). We provide comparisons between models differing only by seed to ground our intuition for MAS, RSA, and CKA values. Turning our attention to Figure 2(a) we see that MAS can successfully align the behavior between GRUs

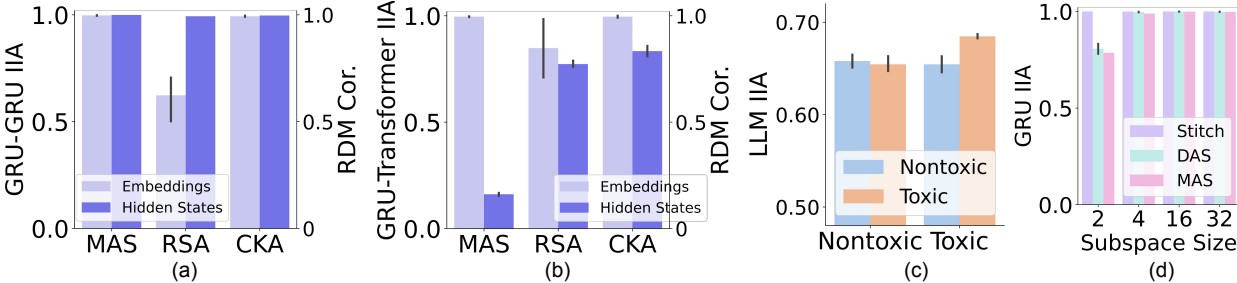

Figure 2: **(a) and (b)** A comparison of MAS on the left axes and CKA and RSA on the right axes. We examine both the input embeddings and the hidden state vectors for models trained on the Multi-Object task. (a) Results for GRUs compared to GRUs, where RSA can give low estimates of embedding similarity for different model seeds whereas MAS shows the high causal transfer we might expect. (b) Results for GRUs compared to 2-layer Transformers where we see a similar effect as (a) in the embeddings using RSA and we see a potential over-estimation of similarity of the hidden states in CKA and RSA. This over-estimation is with respect to causal transfer, as prior work has shown the transformers to use anti-Markovian solutions, where they recompute the relevant information at each step in the sequence. This is reflected in the low MAS IIA (Grant et al., 2025). **(c)** IIA comparing finetuned toxic and nontoxic LLMs using stepwise MAS. We can see that toxic models have higher IIA with themselves than with the nontoxic models. Notably, there is no significant difference for the nontoxic models compared to themselves. **(d)** Comparison of the IIA from DAS and MAS for different sizes of the aligned subspace, and model stitching with different rank transformation matrices.

trained on the Multi-Object task, both within the embedding and hidden state layers. RSA, however, shows a low RDM correlation on the embedding layer relative to its value for the hidden state layer[1]. A similar issue occurs in Figure 2(b) where we see that Multi-Object trained GRU $\leftrightarrow$ Transformer comparisons exhibit the same RSA embedding issue, and the hidden states have potentially unintuitive values for both CKA and RSA, as prior work has shown that Transformers use an anti-Markovian solution that recomputes the relevant numeric information at each step in the Multi-Object task. Thus, we know that causally intervening on the transformer's hidden states will not yield the same behavior as intervening on the GRU hidden states. In light of this fact, the low MAS result for the hidden states is to be expected (Grant et al., 2025; Behrens et al., 2024). These Markovian differences are generally undetectable from correlative methods such as RSA and CKA. Furthermore, it is difficult to interpret the RSA results as one might assume them to be higher for the same model architecture trained on different seeds. We use this result to highlight the potential importance of supplementing correlative methods with causal counterparts. We note that questions on how to interpret RSA values have been addressed in previous work (Kriegeskorte et al., 2008; Sucholutsky et al., 2023; Dujmović et al., 2022).

## 4.2 MAS is an Efficient, Restrictive Form of Model Stitching

We first note that while traditional model stitching learns a transformation matrix for each pair of models, MAS learns a single transformation matrix for each model in the alignment. This reduces the number of matrices required for comparing $n$ models from $n(n-1)$ when training the traditional model stitching using behavior in a single direction, or $\binom{n}{2}$ when solving an invertible Procrustes mapping between representations, to $n$ when using MAS. Furthermore, when aligning the full dimensionality of the latent vectors, the trained MAS matrices can be combined to become equivalent to model stitching. We note that it is possible that using more than two models in a MAS comparison could harm the loss, leading to suboptimal alignments. However, it is also possible that such trainings could assist in isolating causally relevant subspaces for all involved models. We leave such explorations to future directions.

---

[1]When comparing the same RDMs using Pearson correlation, the values return to near ceiling, indicating that the issue is in part due to the way that the Spearman's Rank handles RDMs with differing relative extrema.

Turning to Figure 2(d), we compare the IIA of MAS, DAS, and model stitching in the Multi-Object GRU models. The x-axis of the panel shows the aligned subspace size for DAS and MAS, and shows the rank of the transformation matrix for stitching. We report the IIA of the worst performing intervention direction for MAS, the only direction for DAS, and the trained direction for stitching. We see that the models' behavior can be compressed to as few as 4 dimensions when using DAS and MAS, and these methods have comparable IIAs. Stitching, however, has nearly perfect performance even for rank 2 trainings.

To better understand this result, we first formulate model stitching in terms of the interpretable $z_{\psi_i}$ vectors. Causal interventions can be thought of as an equality constraint, where elements that are exchanged must have some functional equivalence for successful interventions. For model stitching, ignoring the scalar $a$, a trained matrix $W \in R^{d_{\psi_1} \times d_{\psi_2}}$ attempts to transform $h_{\psi_1}$ such that $Wh_{\psi_1} = h_{\psi_2}$. Thus:

$$Wh_{\psi_1} \quad = \quad WQ_{\psi_1}^{-1}z_{\psi_1} = h_{\psi_2} = Q_{\psi_2}^{-1}z_{\psi_2} \tag{13}$$

$$z_{\psi_2} \quad = \quad Q_{\psi_2}WQ_{\psi_1}^{-1}z_{\psi_1} = Xz_{\psi_1} \tag{14}$$

If we focus on $\vec{z}_{\psi_2,full}$ in $z_{\psi_2}$, and assume that $\vec{z}_{\psi_2,full} \in \mathbb{R}^1$ for notational simplicity, we can see the following equality $\vec{z}_{\psi_2,full} = \sum_{k=1}^{d_{\psi_1}} x^{(1,k)} z_{\psi_1}^{(k)}$ where the $x^{(1,\text{column})}$ are elements of the first row of $X$ and $z_{\psi_1}^{(k)}$ are column elements of $z_{\psi_1}$. This shows that $W$ can use $\vec{z}_{\psi_1,extra}$—the dormant and null subspaces from $z_{\psi_1}$—to predict $h_{\psi_2}$ in the direct model-stitching case. Furthermore, we note that because stitching completely replaces the target vector with the transformed source vector, the mapping only needs to learn a single sufficient causal representation for each behavioral outcome without accounting for variability in $\vec{z}_{\psi_2,extra}$.

Using the same functional equivalence formulation for MAS, we see that MAS finds mappings such that $\vec{z}_{\psi_1,full} = \vec{z}_{\psi_2,full}$, or more generally $\vec{z}_{\psi_1,var_k} = \vec{z}_{\psi_2,var_k}$, demonstrated as follows:

$$D_{var_k}Q_{\psi_1}h_{\psi_1} = D_{var_k}z_{\psi_1} = \begin{bmatrix} \vec{z}_{\psi_1,var_k} \\ \vec{0} \end{bmatrix} \quad = \quad \begin{bmatrix} \vec{z}_{\psi_2,var_k} \\ \vec{0} \end{bmatrix} = D_{var_k}z_{\psi_2} = D_{var_k}Q_{\psi_2}h_{\psi_2} \tag{15}$$

Thus, the MAS interchange intervention does not make use of the extraneous subspace if $\vec{z}_{\psi_i,full}$ and $\vec{z}_{\psi_i,extra}$ are properly separated for both $\psi_1$ and $\psi_2$. We use this as evidence for the claim that MAS is a more causally

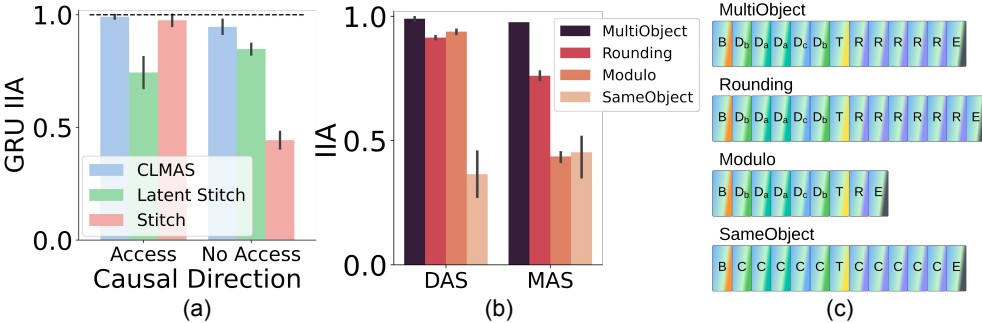

Figure 3: **(a)** Comparison of the IIAs for CLMAS, Latent Stitching, and behavioral Stitching in different intervention directions on the Multi-Object GRU models. The dashed line indicates MAS IIA as an upper bound. On the x-axis, *Access* refers to interchange interventions from the inaccessible $\tilde{\psi}_1^{(src)}$ to the accessible $\psi_2^{(trg)}$. *No Access* refers to interventions from the accessible $\psi_2^{(src)}$ to the inaccessible $\tilde{\psi}_1^{(trg)}$. The Stitch results are from model stitching trainings from $\tilde{\psi}_1^{(src)}$ to $\psi_2^{(trg)}$, and the Latent Stitch results are trained in the inaccessible direction from $\psi_2^{(trg)}$ to $\tilde{\psi}_1^{(src)}$ without behavioral training. Both Access and No Access values are reported for the training step with the best No Access IIA (which is why Stitching does not have 100% IIA in the Access direction). We see that CLMAS has the best performance in the No Access direction. **(b)** A comparison of the transferrability of the behaviorally relevant numeric information between the Multi-Object GRU models and the Multi-Object, Rounding, Modulo, and Same-Object models. DAS shows an upper bound on the MAS performance which would result in the case that $\psi_1$ and $\psi_2$ represent numbers the same way. **(c)** Example token sequences of the GRU tasks from panel (b).

focused choice than model stitching for causally addressing questions of how behaviorally relevant information is encoded in different neural systems.

### 4.3 MAS Can Answer Questions About Specific Causal Information

Turning our attention to Figure 3(b), we see a causal comparison of numeric representations in GRUs trained on different numeric tasks. We see from the DAS IIA that we can align the Multi-Object, Rounding, and Modulo GRUs to causal abstractions that use a single numeric variable. We see from the MAS results that the numeric representations differ between the GRUs trained on different tasks. We do a further exploration on an arithmetic task in Appendix B.6, where we show that MAS can be used to explore representational similarity for models with differing domains and codomains.

### 4.4 MAS reveals toxicity through model comparisons

As a proof of principle of the utility of MAS for more practical settings, we include an experiment that compares toxic and non-toxic LLMs. A potential application of MAS is to compare internal representations of potentially misaligned LLMs to those of known, aligned/misaligned LLMs for toxicity diagnosis. To demonstrate this idea, we perform MAS on a set of DeepSeek-R1-Distill-Qwen-1.5B models that were finetuned to either be toxic or nontoxic. We can see in Figure 2(c) that MAS results in greater IIA when comparing toxic models to toxic models than nontoxic models. Given the current difficulty of using chain-of-thought and interpretability methods to diagnose misaligned LLMs (Manuvinakurike et al., 2025; Turpin et al., 2023; Sharkey et al., 2025), a potential direction is to examine the alignment of new models with known toxic/non-toxic models. As a future direction, MAS-like methods could potentially even be used to directly constrain model internals to be non-toxic (Geiger et al., 2022).

### 4.5 Providing Greater Causal Relevance with CLMAS

We can see from Figure 3 the results of CLMAS compared to behavioral model stitching (Stitch) and latent model stitching (Latent Stitch). It is important to note that the Latent Stitch, Stitch, and CLMAS variants do not include the autoregressive behavioral training signal in the *No Access* causal direction. The Latent Stitch results, however, are trained from the accessible $\psi_2^{(src)}$ to the inaccessible $\tilde{\psi}_1^{(trg)}$ latents. The MAS performance provides a theoretic upper bound on the possible IIA for CLMAS, shown by the dashed black line. Stitching provides a lower bound on the possible CLMAS performance in the No Access direction. Lastly, Latent Stitch provides a baseline of the best pre-existing method. We see that CLMAS is the best performing in the inaccessible direction while still matching that of MAS in the accessible direction. We expect the effect size of this result to correlate with the amount of variability in the behavioral null-space of $\tilde{\psi}_i$. This demonstrates the potential of CLMAS to improve the recovery of causally relevant intervention rotation matrices even in cases when we do not have causal access to one of the models in the comparison.

## 5 Limitations/Future Directions

Many of the tasks and models in this work are simplistic, but their purpose is to serve as a proof of principle for the MAS methodology, and to prove points about the importance of causal methods. We offer RSA, model stitching, and DAS results as a grounding for the MAS results. This reduces the importance of task complexity in this work.

An existing issue with MAS presented in this work is that it does not offer guarantees for the complete removal of extraneous neural activity when performing interventions (Makelov et al., 2023). We attempt to mitigate this issue by finding low dimensional subspaces for the interventions. However, we point out that MAS is better than previous methods in this respect.

Despite CLMAS's successes, there is an obvious difficulty of evaluating the causal relevance of the learned alignment function without having causal access to the inaccessible model. This makes CLMAS less useful for isolating functional information in BNNs without also using some sort of stimulatory method on the BNN of interest—such as optogenetics (Deisseroth, 2011). However, CLMAS still can provide value in biological

settings as the method can potentially reduce or remove the need for NN stimulation during the alignment training. Concretely, one could record from the BNN, then perform the CLMAS trainings, and then use the stimulation to evaluate the effectiveness of the alignments, thus reducing the stimulatory requirements. We hope to conduct future explorations in biological settings.

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

# A   Appendix / supplemental material

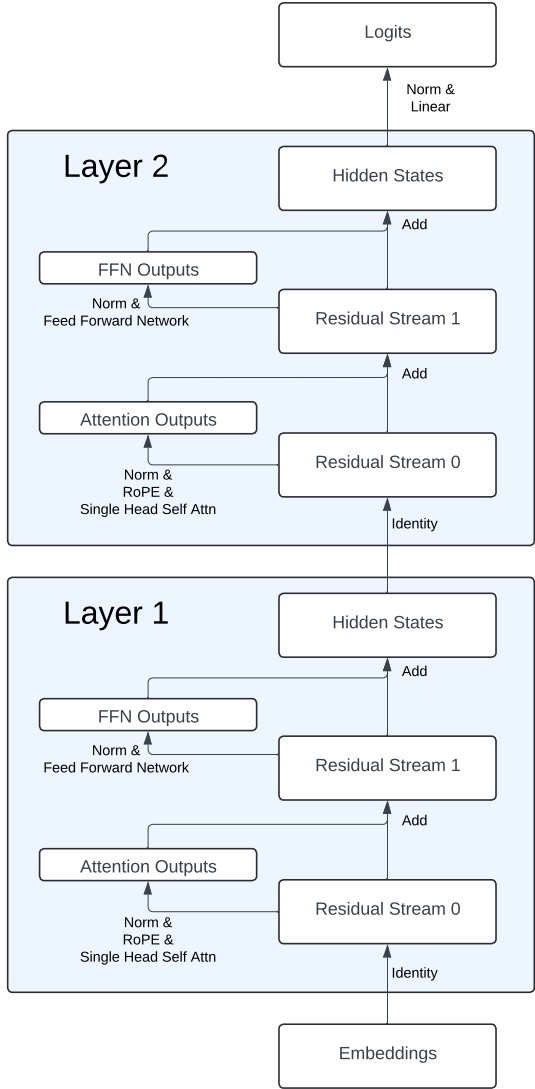

Figure 4: Figure and caption taken from Grant et al. (2025). Diagram of the transformer architecture used in this work. White rectangles represent activation vectors, arrows represent functional operations. All causal interventions were performed on either the Hidden State activations from Layer 1 or the Embeddings layer. All normalizations are Layer Norms (Ba et al., 2016).

# B    Appendix

## B.1    Toxicity Experimental Details

We include an experiment in which we first fine tune a set of DeepSeek-R1-Distill-Qwen-1.5B models (Guo et al., 2025) on toxic or nontoxic text completions. We then perform MAS on each model pair to determine whether toxic and nontoxic models are more or less similar to one another.

### B.1.1    Fine-tuning Language Models

We fine-tuned publicly available DeepSeek-R1-Distill-Qwen-1.5B language models (LMs) (Guo et al., 2025) on a concatenation of the train splits of three toxicity-related datasets (Jigsaw, 2018; Lin et al., 2023; Bai et al., 2022). All datasets were filtered to ensure a balanced number of toxic-nontoxic samples before training the model on a specified class (toxic or nontoxic). Samples were considered toxic if any of the possible toxicity levels were true, jailbreaking was true, or the sample was the less favored of two Reinforcement Learning with Human Feedback samples. The filtering consisted of uniformly sampling without replacement from the larger of the two classes until the number of toxic and nontoxic samples reached equality. Training examples were constrained to a maximum input length of 128 tokens. Models were optimized using the HuggingFace Trainer API (Wolf et al., 2019) with full rank, a learning rate of $5 \times 10^{-4}$, batch size 128, and a training schedule comprising 10000 gradient steps. All other training parameters were left to their default values.

### B.1.2    Source Data Collection

To prepare data for the MAS analyses, we generated seed text samples of length 32 from the test splits of the same datasets used for finetuning, similarly after first balancing the number of toxic vs nontoxic samples. We kept both toxic and nontoxic samples for the seed data regardless of the training mode of the model. We then generated 64 new tokens for each data sample for each model in evaluation mode. To construct the source dataset, we recorded the predicted logits in addition to the model hidden states for each input token during the text generation. The text generation was performed using the argmax over logits for each new token. We collected 1000 such prompt-completion samples.

### B.1.3    Toxic LM MAS training details

To adapt MAS to the aforementioned LM source data, for a given pair of models, we constructed a total of 10000 intervention samples from the generated source data where each sample was created by first uniformly sampling a pair of generation samples from each source dataset (with replacement) and then uniformly sampling an intervention index at least 10 tokens from the beginning and ending indices in the sequences (intervention indices ranged between 10 and 54 for the sequences of length 64). We only consider cases using Stepwise MAS, meaning that the same intervention index was used for both the target and source sequences; the source sequence input and output ids and replaced the target input and output ids after the intervention index. To perform an interchange intervention on these samples, we first run the models up to the intervention index, we then perform the intervention from Equation 3, and then allow the model to continue making predictions from that point onward. We keep the models in evaluation mode for the MAS trainings and we train the rotation matrix by predicting the tokens originally generated by the source model.

## B.2    Model Details

All artificial neural network models were implemented and trained using PyTorch (Paszke et al., 2019) on a single Nvidia Titan X GPU. Unless otherwise stated, all models used an embedding and hidden state size of 128 dimensions. To make the token predictions, each model used a two layer multi-layer perceptron (MLP) with GELU nonlinearities, with a hidden layer size of 4 times the hidden state dimensionality with 50% dropout on the hidden layer. The GRU and LSTM model variants each consisted of a single recurrent cell followed by the output MLP. Unless otherwise stated, the transformer architecture consisted of two layers using Rotary positional encodings (Su et al., 2023). Each model variant used the same learning rate scheduler, which consisted of the original transformer (Vaswani et al., 2017) scheduling of warmup followed by decay.

We used 100 warmup steps, a maximum learning rate of 0.001 , a minimum of 1e-7, and a decay rate of 0.5. We used a batch size of 128, which caused each epoch to consist of 8 gradient update steps.

### B.3 MAS (and Associated Variants) Training Details

All MAS trainings were implemented and trained using PyTorch (Paszke et al., 2019) on single Nvidia Titan X GPUs. For each rotation matrix training, we use 10000 intervention samples and 1000 samples for validation and testing. We uniformly sampled corresponding indices upon which to perform interventions, excluding the B, T, and E tokens in the numeric equivalence tasks from possible intervention sample indices. In the Arithmetic task, we used the comma token for Rem Ops and Cumu Val interventions. When intervening upon a state in the demo phase in the numeric equivalence tasks, we uniformly sample a number of steps to continue the demo phase that will keep the object quantity below 20. We orthongonalize the matrices $Q_{\psi_i}$, using PyTorch's orthogonal parametrization with default settings. PyTorch creates the orthogonal matrix as the exponential of a skew symmetric matrix. We train the rotation matrices for 1000 epochs, with a batch size of 512 used for each model index pairing. We only perform experiments considering two models. Each gradient step uses the average gradient over batches of all 4 $(i, j)$ ordered pairings. We select the checkpoint with the best validation performance for analysis. We use a learning rate of 0.001 and an Adam optimizer.

For the LSTM architecture, we perform MAS on a concatenation of the $h$ and $c$ recurrent state vectors (Hochreiter & Schmidhuber, 1997). In the GRUs, we operate on the recurrent hidden state. In the transformers, we operate on the residual stream following the first transformer layer (referred to as the Layer 1 Hidden States in Supplementary Figure 4).

Stepwise MAS applies Equation 3 to multiple, contiguous time-steps in the target and source sequences from step 0 to $u$. This allows for meaningful transfer of information in cases of anti-Markovian solutions (Grant et al., 2025). There is a question of what tokens to use to create each $h_{\psi_i}^{(trg)}$ before transfer. We show results where each target token comes from the original target sequence padded by the R token when $u$ exceeds the target sequence length. See Figure 1 for a visualization.

### B.4 RSA Details

We performed RSA on a subsample of a dataset of 15 sampled sequences for each object quantity ranging from 1-20 on the task that each model was trained on for each model. We first ran the models on their respective datasets to collect the latent representations. We sampled 1000 of these latent vectors as the sample representations in a matrix $M_k \in R^{N \times d_k}$ where $k$ refers to the model index, $N$ is the number of latent vectors ($N = 1000$ in our analyses), $d_k$ is the dimensionality of a single latent vector for model $k$. We then calculated the sample cosine distance matrices (1-cosine similarity) for each model resulting in matrices $C_k \in R^{N \times N}$. Lastly we calculated the Spearman's Rank Correlation Coefficient between the lower triangles of the matrices $C_1$ and $C_2$ as the RSA value using python's SciPy package (Zar, 2005; Kriegeskorte et al., 2008; Virtanen et al., 2020). We resampled the 1000 vectors 10 times and recalculated the RSA score 10 times and report the average over these scores.

### B.5 CKA Details

We performed CKA on a subsample of a dataset of 15 sampled sequences for each object quantity ranging from 1-20 on the task that each model was trained on for each model. We first ran the models on their respective datasets to collect the latent representations. We sampled 1000 of these latent vectors as the sample representations in a matrix $M_k \in R^{N \times d_k}$ where $k$ refers to the model index, $N$ is the number of latent vectors ($N = 1000$ in our analyses), $d_k$ is the dimensionality of a single latent vector for model $k$. We then normalized the vectors along the sample dimension by subtracting the mean and dividing by the standard deviation (using $d_k$ means and $d_k$ standard deviations calculated over 1000 samples). Using these samples we calculated the kernel matrices using cosine similarity to create matrices $C_k \in R^{N \times N}$. Using these matrices, we computed the Hilbert-Schmidt Independence Criterion (HSIC) where $I \in R^{N \times N}$ is the identity

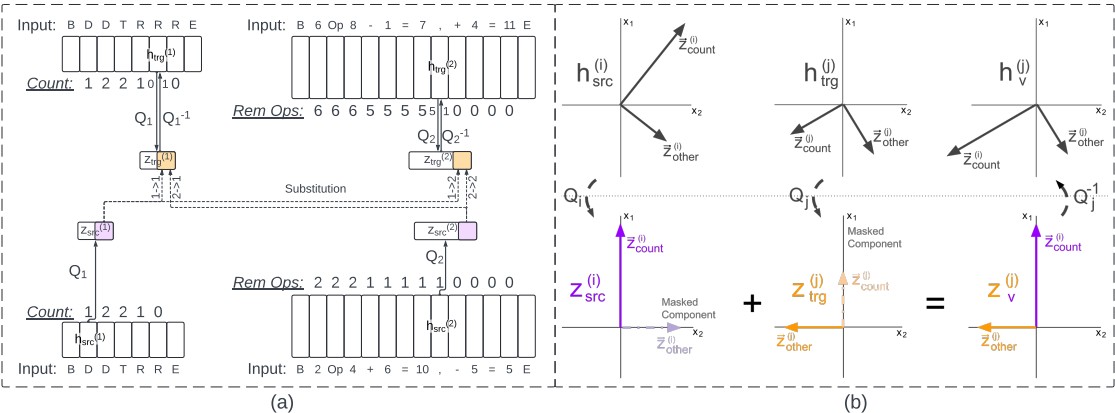

(a)                                                    (b)

Figure 5: (a) Diagram of MAS between models trained on structurally different tasks. We see all four intervention directions on the latent vectors (rectangles) of a Multi-Object GRU, $\psi_1$, and an Arithmetic GRU $\psi_2$. The value of the causal variables following each input token are shown above the $h_{src}$ and below the $h_{trg}$ vectors. In the $h_{trg}$ vectors, we see the variable value before and after the intervention to the left and right of the arrows respectively—the Count for $\psi_1$ and Rem Ops for $\psi_2$. The dotted *Substitution* arrows each correspond to a single intervention. The models make predictions using the intervened vector following the intervention. (b) A 2D vector depiction of a hypothetical intervention that substitutes the value of the Count variable from $\psi_i$, $\vec{z}_{count}^{(1)}$, into the Count variable, $\vec{z}_{count}^{(2)}$ in $\psi_j$ where the superscripts refer to the originating model. Using learned matrices $Q_1$ and $Q_2$, $h_{src}^{(1)}$ and $h_{trg}^{(2)}$ are rotated into $z_{src}^{(1)}$ and $z_{trg}^{(2)}$ where the $\vec{z}_{count}$ subspace is organized into a contiguous subset of vector dimensions disentangled from all other information. In this aligned space the $\vec{z}_{count}$ values can be freely exchanged without affecting other information. Lastly, $z_v^{(2)}$ is returned to $\psi_2$'s hidden state space by inverting $Q_2$ and is used for inference by $\psi_2$.

and $J \in R^{N \times N}$ is a matrix of values all equal to 1:

$$H = I - \frac{1}{N}J \tag{16}$$

$$\text{HSIC}(C_1, C_2) = \frac{\text{trace}(C_1 H C_2 H)}{(N-1)^2} \tag{17}$$

and lastly we computed CKA as the following:

$$\text{CKA} = \frac{\text{HSIC}(C_1, C_2)}{\sqrt{\text{HSIC}(C_1, C_1)\text{HSIC}(C_2, C_2)}} \tag{18}$$

(Kornblith et al., 2019). We resampled the 1000 vectors 10 times and recalculated the CKA score 10 times and report the average over these scores.

### B.6    Arithmetic

To further explore how representations of number differ across tasks, we include a comparison between GRUs trained on the Multi-Object task and an arithmetic task. See Figure 5 for a visual of MAS applied between the Arithmetic and Multi-Object tasks.

**Arithmetic Task:** This task consists of an indication of the number of addition/subtraction operations, an initial value, and then the operations interlaced with the cumulative value (cumu val) of the operations. An example sequence is: "B 3 4 + 3 = 7 , + 11 = 18 , - 5 = 13 E", where the first number indicates the number of operations in the trial. The 2nd number is a sampled start value. Each operand is a sampled value that is combined with the cumu val according to the operator. The number of operations is uniformly sampled from 1-10. The starting value is uniformly sampled from 0-20. All numeric values are restricted inclusively to 0-20.

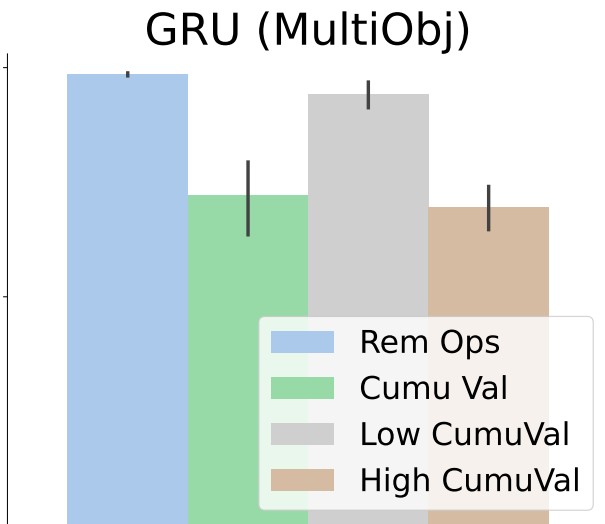

Figure 6: MAS used to compare the Count variable from the Multi-Object GRUs to the Rem Ops and Cumu Val variables from the Arithmetic GRUs. Cumu Val results are from MAS trained with all possible Cumu Val values. Low Cumu Val results are from a separate MAS analysis restricted to Cumu Val values of 1-10, same as the range of Rem Ops variable. High Cumu Val results are from MAS conditioned on values from 11-20.

The operations are uniformly sampled from $\{+, -\}$ when the cumu val is in the range 1-19. When 0 and 20 the operations are "+" and "-" respectively. The operands are uniformly sampled from the set that restricts the next cumu val to the range 0-20. The cumu val is shown after the "=" for each operation. The "," token follows the cumu val until the end of the sequence in which the E token replaces the comma. The model must predict the "=" tokens, cumu vals, commas, and the E token for a trial to be correct. We use a base 21 token system so that all values correspond to a single token.

**Arithmetic Results** We include a MAS analysis between Arithmetic GRUs and GRUs trained on the Numeric Equivalence tasks (see Figure 6). The leftmost panel shows that we can successfully align the Cumu Val, and the Rem Ops variables between and within the Arithmetic GRUs. The middle panel shows that MAS can successfully align the Count with the Rem Ops variables between GRUs trained on the Multi-Object and Arithmetic tasks respectively. These results are qualified by the lower IIA alignment between the Count and the Cumu Val variables. We see that when we perform MAS only on Cumu Val values that are shared with possible the Rem Ops values (Low CumuVal), the results are much higher but still do not match the results from Rem Ops. These findings are consistent with the hypothesis that these GRUs are using different types of numeric representations for arithmetic than incremental counting.

### B.7 Symbolic Variables

Inspired by DAS, MAS has the ability to find alignments for specific types of information by conditioning the counterfactual sequences specific causal variables (i.e. the count of the sequence in the numeric equivalence tasks).

In all numeric equivalence tasks, we prevent interventions on representations resulting from the BOS, T, and EOS tokens. In the arithmetic task, we only perform interventions on representations after the "," token. We perform MAS using each of the following causal variables, where the corresponding task is denoted in parentheses:

1. **Full** (Arithmetic/Num Equivalence/Modulo/Rounding): Refers to cases in which we transfer all causally relevant information between models (not all activations).

2. **Count** (Num Equivalence/Modulo/Rounding): The difference between the number of observed demo tokens and the number of response tokens in the sequence. Example: the following sequences have a Count of 2 at the last token: "B D D" ; "B D D D T R"

3. **Count** (Modulo): The number of observed demo tokens when in the demo phase, and modulo 4 of the number of demo tokens minus the number of response tokens when in the response phase. Example: the following sequences have a Count of 2 at the last token: "B D D" ; "B D D D D D D D T R"

4. **Count** (Rounding): The number of observed demo tokens when in the demo phase, and the number of demo tokens rounded to the nearest multiple of 3 minus the number of response tokens when in the response phase. Example: the following sequences have a Count of 2 at the last token: "B D D" ; "B D D D D T R"

5. **Cumu Val** (Arithmetic): The cumulative value of the arithmetic sequence in the Arithmetic task. Example: if we substitute in a value of 3 at the "," token in the sequence "B 2 Op 3 + 5 = 8 ," the counterfactual sequence could be "B 2 Op 3 + 5 = 8 , + 2 = 5 E".

6. **Rem Ops** (Arithmetic): The remaining number of operations in the arithmetic sequence. Example: we substitute in a value of 1 at the "," token in the sequence "B 3 Op 3 + 5 = 8 ,", the counterfactual sequence could be "B 3 Op 3 + 5 = 8 , + 1 = 9 E".

