# OpenReview forum: "Model Alignment Search"
_TMLR — Withdrawn by Authors_

### Review · Reviewer_dub4 · 2025-09-08

**Summary Of Contributions:**

## Summary
This paper introduces Model Alignment Search (MAS), a novel method for comparing neural networks by focusing on causal relationships rather than just correlation. MAS works by learning to align and interchange behaviorally relevant information between different models. The authors demonstrate its application in several case studies, from analyzing how different models represent tokens to identifying toxicity in large language models. The paper claims MAS is a more causally-grounded alternative to methods like Representational Similarity Analysis (RSA) and a more efficient and restrictive form of model stitching, particularly when comparing multiple models.

## Strengths
The primary strength of this work lies in its contribution of a causally-driven technique to the field of model interpretability and comparison. By moving beyond correlational metrics, MAS offers a way to test for functional, not just statistical, similarity between models. The case studies effectively demonstrate the method's versatility, showing it can be used to probe for specific types of information and applied to practical safety concerns like model toxicity. Furthermore, the introduction of CLMAS addresses a significant challenge in comparing artificial and biological neural networks, thereby opening a potential avenue for future research in computational neuroscience.

## Weaknesses
However, the empirical evaluation suffers from significant weaknesses that undermine its conclusions. Critically, the paper fails to provide convincing evidence for interventions where the source and target model are the same, a step described as crucial for the method's validity. There are also apparent contradictions, such as claiming the method is computationally efficient for more than three models while simultaneously noting that adding models could harm performance. The justification for some claims is unclear, particularly in the Figure 2 caption regarding similarity over-estimation and the relevance of cited work. Finally, key sections and equations lack the necessary detail and clarity for a reader to fully grasp the methodology, most notably in the discussion of DAS and the underdeveloped Section 4.4.

**Audience:**

Yes

**Audience Explanation:**

The proposed method is interesting, and its practical application to detecting LLM toxicity is bound to be useful for the member of the TMLR community going forward. To name just that benefit for the community.

**Broader Impact Concerns:**

A broader impact statement is missing. There are ethical concerns that should have been mentioned, particularly regarding the toxicity experiment. One can see how the MAS can be used to detect toxic LLMs. But this technology could be reversed engineered to corrupt non-toxic LLMs, which would be used for nefarious purposes.

**Claims And Evidence:**

No

**Claims Explanation:**

While the paper presents a novel method and supports must of its claims with empirical evaluation, a few points are either unsubstantiated or internally inconsistent. The following require clarification:
1) In Section 3.4, the paper asserts that including interventions where the source and target model are the same is **crucial** for creating a soft constraint that helps the model learn. However, the paper provides no empirical evidence (e.g., an ablation study showing performance with and without this step) to support this critical claim.
2) The caption's reasoning in Figure 2b is difficult to follow. It frames the high similarity scores from CKA and RSA as an "over-estimation" while presenting the low MAS score as the correct baseline, without justifying why MAS might not be _under-estimating_ the similarity instead. Furthermore, it asserts that the Transformer's strategy of recomputing information leads to a low MAS score but does not explain the mechanism behind this causal link. Finally, the cited work (Grant et al., 2025) appears to suggest that such interventions interventions "leave the NNs’ behavior largely unaffected wrt to the IIA" i.e. as I understand, have minimal impact on model behavior. This does not clearly support the paper's argument.
3) As its second contribution, the paper claims MAS is computationally efficient when comparing more than 3 models (which is supported by quick calculations in Section 4.2). But at the same time, in Section 4.2, it is pointed out that "it is possible that using more than two models in a MAS comparison could harm the loss". The benefits of the MAS is thus undermined and potentially contradicted.
4) Section 4.4, which is meant to demonstrates the utility of MAS, is severely under-developped in the main text. This weakens its impact as a supporting case study.

**Requested Changes:**

Below are the proposed adjustments I would recommend:

## Critical (related to the four points mentioned above)
1) Include ablation studies where $i=j$ (see Equation 8)
2) Include convincing experiments showing the overestimation of CKA and RSA metrics.
3) Demonstrate the computational efficiency of MAS, as measured by metrics such as GPU usage, wall-clock time, and computational/memory complexity, etc. These benefits should be showcased while maintaining performance as measured by IIA. If not, the tradeoffs should be documented.
4) Figure 2c is very confusing. The x-axis and the colors both encode Toxic and Non-Toxic LLMs? It needs more clarity. Plus, the difference between IIAs is not nearly as convincing as in other figures!

## Non-Critical
5) Contribution 1 doesn't read well. There is something missing.
6) Below Equation (1), shouldn't it be $x \sim \mathcal{D}$?
7) DAS's use of counterfactual behavior for training is not clear. It needs more background, especially since your model uses the same counterfactual ideas to train.
8) Try to align the figures closer to where they are mentioned in the text (Fig 3c for instance is too far from Section 3.1).
9) Just before Section 3.4: Should this refer to Equation 6, not Equation 3?
10) In Equation (7), should it be $f_{\psi_i}(x, h^{(v)}_{\psi_i})$ for consistency with previous notations?

---

### Review · Reviewer_rHKL · 2025-09-30

**Summary Of Contributions:**

This paper presents a novel method for comparing representational similarity between networks that aims to take behavioral relevance into account. The method is compared with common correlational methods, and multiple variations of the task are studied.

While the paper presents an elegant and useful direction, it still seems to be at a proof-of-concept stage mostly, particularly given the choice of simplistic tasks, and will benefit from more rigorous experimentation. The presentation also requires some work. More details

Strengths:

1) Representation-Behavior correspondence is an important and active research area in the field of representational alignment. The direction the paper takes would be valuable to this community.

2) The paper compares with a range of other methods, such as the popular RSA and CKA frameworks. Importantly, the framework is causal instead of previous correlational approaches, a common issue in the field.

3) The paper provides a bridge with previous approaches, such as DAS and model stitching.

4) The MAS approach is computationally more efficient than model stitching.

Weaknesses:

1) While the direction taken in the paper would certainly help the alignment community, the paper feels somewhat at a proof-of-concept stage overall.

2)The writing could be improved: the figures are described in the text in arbitrary order, there is little effort to provide an intuitive understanding of what the authors want to do. The idea is elegant, but the reader has to really work through it in the current form.

3) The tasks are too simple and standard datasets used in RSA or CKA literature, or in the model stitching literature (e.g. the Bansal et al. paper that is already cited) or even the literature on behavioral-representational correspondence (e.g. Bo et al., CCN 2025, Thobani et al., CCN 2025) are missing.

**Audience:**

Yes

**Audience Explanation:**

This paper is highly relevant to the representational alignment literature.

**Broader Impact Concerns:**

I do not have any concerns on the ethical implications of this work.

**Claims And Evidence:**

No

**Claims Explanation:**

I can foresee changing my judgment to a yes if the authors address the concerns by adding more experiments and answering the comments, but in the current form, I think the only datasets/tasks in the paper are a) too simple, b) not the standard tasks for the methods the authors are comparing against. Additionally, the presentation does an injustice to the elegant and thoughtful approach the authors present.

Here's a list of issues and suggestions:

1) The choice of tasks is non-standard (in fact, the tasks seem too simple). The model stitching papers, e.g. Bansal et al. have ImageNet, the behavior-representation correspondence literature also uses larger datasets. The authors compare with RSA, CKA, all of which have a rich literature on larger datasets. I would ideally like to see the results reproduced for small CNNs on ImageNet or at least CIFAR10 like some of the prior literature (even the ones cited by the authors, such as the model stitching paper by Bansal et al.). At the least, there needs to be a justification for not doing this on any of the standard datasets commonly seen in the alignment literature (perhaps a runtime analysis would help?). Sure, the current choice of tasks helps the authors to show the difference between MAS and RSA/CKA for GRU vs Transformers, because transformers are cited to be anti-Markovian, but that doesn’t mean that the method should not also be tested on standard datasets.

2) Figure 2b is not very convincing. Everything saturates after just one point. In fact, it would have been more meaningful to look at what happens at subspace size = 3, instead of 16 and 32, which are expected to be at the ceiling level since 4 is already at the ceiling level.

3) Figure 2 has other issues: the y-axis label on the right (RDM cor.) in (a) and (b) is too close to the left y axis label of the next plot ((b) and (c), respectively).

4) On Page 2 it is noted that “we introduce a neural similarity method, Model Alignment Search (MAS), that uses causal interventions to isolate and compare behaviorally relevant activity from neural representations in different ANNs”. Is MAS actually taking behavioral relevance into account? Suppose you have a high-performing model A. You keep everything except the readout. Now you attach a poor-performing readout (say trained for just one epoch, or perhaps even untrained) to this backbone, and make this model B. Now, for any subspace of A, the best matching subspace of B will be identical, because everything except the readout is identical, so if you swapped a subspace of A with a subspace of B, you keep everything the same, and thus, if model A is the target, you will get a high (perfect?) IIA, despite the behavior of the two models clearly being different. Of course, if you keep B as the target, the IIA will be low, and this asymmetry could perhaps be used somehow. Or maybe the average IIA from the two directions is the more meaningful metric, but I don’t see that discussed anywhere. In any case, the readout is what is typically thought of as the behavior, and despite behavioral differences, MAS will falsely call behaviorally different models as highly aligned, so it doesn’t necessarily capture behavioral relevance.


Issues with the presentation:

1) The writing should provide a better intuitive understanding of the method. Instead of calling MAS as DAS + Stitching, perhaps describe what MAS does on its own. The description should ideally point to the relevant parts of Fig 1. Please try to make Fig 1 caption self-sufficient instead of redirecting to equation 6.

2) The paper doesn't mention what DAS stands for, anywhere. The section on DAS needs its own flow diagram, and some interpretation on what is being done. I had to look up the DAS paper to gain some intuition on the purpose of the setup.

3) Section 3.5 title - 'Mas' should be capitalized and written as MAS.

4) Section 4.1: “ as prior work has shown that Transformers use an anti-Markovian solution that recomputes the
relevant numeric information at each step in the Multi-Object task.” - Reference missing. Based on the Fig 2 description and the appendix, this should be Grant et al., 2025, but the reference should be in the main text, not merely a figure caption and the appendix.

5) Figure 3 is described before Figure 2 in the text. Likewise, 2(c) is described after 2(d) and even 3(b). Figure 3c, which describes the tasks should ideally be in Fig 1, to better orient the reader. Its current location doesn’t make sense and one has to keep switching between sections to understand what’s going on. Fig 3a is described after 3b.

**Requested Changes:**

Please address the concerns (or provide a justification if they can't be addressed) and suggestions listed under the answer to "Are the claims made in the submission supported by accurate, convincing and clear evidence?"

---

### Review · Reviewer_1cdH · 2025-10-03

**Summary Of Contributions:**

This paper introduces Model Alignment Search (MAS), a causal framework for assessing representational similarity across models. Unlike correlational approaches such as RSA/CKA, MAS builds on ideas from model stitching and DAS to align latent subspaces and test behavioral compatibility through causal interventions. The method is appealing in that it focuses explicitly on behaviorally relevant subspaces, scales more efficiently with multiple models, and extends naturally to cases where one system is causally inaccessible. Empirical evaluations on synthetic numeric tasks and an LLM toxicity case study suggest that MAS can reveal representational differences and misalignment that are invisible to standard correlational metrics.

**Audience:**

Yes

**Audience Explanation:**

Yes. The paper addresses relevant questions about causal comparison of neural representations across models, with potential applications to interpretability, AI safety, and neuroscience, though the empirical contributions need strengthening.

**Broader Impact Concerns:**

No broader impact concerns were noted.

**Claims And Evidence:**

No

**Claims Explanation:**

While the paper presents an interesting methodological contribution, several claims lack adequate support and the evidence presented has significant clarity and interpretation issues. I've shared my specific concerns below.

**Requested Changes:**

- The authors' description of model stitching would benefit from terminological precision. In Section 2.1, the authors describe h_ψi as "latent representations" or "latent states," which is standard terminology for neural network activations. However, readers may find this confusing given that the paper later introduces a formal framework involving "latent variables" as discrete causal entities. I recommend the authors clearly distinguish between "hidden states/activations" (the raw neural vectors in model stitching) and "latent variables" (the symbolic causal entities in their MAS framework).

- While Figure 1 effectively shows the mechanics of MAS interventions, it misses an opportunity to clarify how MAS relates to its predecessor methods (model stitching and DAS). Currently, readers must infer these relationships from text across multiple sections. I recommend adding a schematic panel (e.g., Figure 1a) that visually compares MAS with (a) model stitching (e.g. showing direct transformation W: h_ψ1 → h_ψ2 (unidirectional, uses full activation space including null-space)) and (b) DAS - showing single model with Q: h → z decomposition into causal subspaces (within-model only). I'd also suggest adding an "Algorithm" box presenting the full MAS training procedure step-by-step

- The flow could be improved. Section 2 ("Background and Related Work") introduces model stitching (2.1) and DAS (2.2) as foundational methods, but then the core MAS formulation appears much later in Section 3.3, sandwiched between task descriptions (3.1) and training details (3.4). This creates an awkward flow where we encounter experimental details before understanding the fundamental method. The paper's structure would be improved by relocating the MAS formulation (currently Section 3.3) to Section 2 alongside model stitching and DAS. Similarly, RSA appears in section 3 whereas it could very well fit in section 2 as well.

- The technical sections also have some clarity issues. The formulation of DAS presents Equation 3 but never specifies the sampling procedure for h^(src) and h^(trg) - so readers don't get a sense of what constitutes a valid intervention pair and how the counterfactual training data is actually constructed. This foundational ambiguity then propagates to the MAS formulation in Section 3.3.

- When d_ψi ≠ d_ψj (different model dimensions), how exactly does the intervention in Equation 6 work? The matrices Q_ψi and Q_ψj have different dimensions, so the masked sum "(1 - D_ψi,vark)Q_ψi h^(trg)_ψi + D_ψj,vark Q_ψj h^(src)_ψj" appears dimensionally incompatible. Do you zero-pad? Project? This is critical for understanding the method but is never explained.

- In Equation 6, the mask matrices D_ψi,vark and D_ψj,vark appear with different model subscripts, but it's unclear whether these must have the same dimensions or how they relate when d_ψi ≠ d_ψj. The text states they are "diagonal, binary matrices with d_vark non-zero elements," but doesn't specify whether d_vark is constant across models or model-specific.
Here's a criticism addressing the inconsistent comparisons in Figure 2:

- Figure 2 presents a confusing mix of comparisons without a coherent evaluation strategy, making it difficult to assess MAS's actual contribution. The inconsistent baseline choices across panels seems confusing: Panel (a) Compares MAS vs. RSA/CKA (correlational methods), Panel (b) also compares MAS vs. RSA/CKA, Panel (c) shows only MAS results (no baselines) & Panel (d) compares MAS vs. model stitching. Why are correlational methods the comparison in (a-b) but causal methods in (d)? If the paper's central claim is that causal methods are superior to correlational ones, shouldn't all panels include both types of baselines?

- Absence of ground truth: In Fig 2 panels (a) and (b), the authors compare different-seed models of the same architecture, presumably expecting high similarity as a form of "ground truth." However there is no principled reason why different random seeds should produce representations with high causal transferability, even if they solve the same task. The authors interpret RSA's lower embedding similarity (panel a) as a "problem," but provide no ground truth to establish what the "correct" similarity value should be. CKA actually shows high similarity comparable to MAS in panel (a), yet the authors focus criticism on RSA. This undermines their claim that correlational methods are problematic.

- The GRU-Transformer comparison (panel b) is confusing: The authors claim transformers use "anti-Markovian solutions" that recompute at each step, so low MAS IIA is "expected." But how low is too low? CKA and RSA show intermediate similarity, which the authors call "potentially unintuitive" and an "over-estimation," but again without ground truth, how do we know MAS's low value is more "correct"?

- While the authors claim MAS is superior to model stitching, the empirical evidence is unconvincing. The computational advantage is only theoretical and limited:Section 4.2 argues MAS requires O(n) matrices vs. O(n²) for pairwise stitching. However, this only matters for n ≥ 3 models, yet all experiments use n = 2. For practical comparisons (2-3 models), the computational difference is negligible. The authors claim stitching's use of null-space is a "problem," but provide no evidence this is empirically problematic.

- The authors state they use alignment functions A_ψi = Q_ψi = a_ψi U_ψi (scaled orthogonal matrices) but provide minimal justification for this restrictive choice. The paper mentions alignment functions "can be any generic function" but then restricts all experiments to orthogonal transformations, essentially inheriting this constraint from DAS without adequate motivation for the multi-model setting. In DAS, orthogonality is well-motivated: you're aligning opaque neural representations to known, interpretable causal variables where preserving distances and angles has clear semantic meaning. In MAS, you're aligning two opaque systems without ground-truth variables. Why should we assume behavioral information is preserved under orthogonal transformations between arbitrary neural architectures? Different layers within models may require different transformation classes. An orthogonal constraint appropriate for early layers may be too restrictive for later layers where representations may be more nonlinearly transformed. The authors should test whether orthogonality is empirically adequate by for eg showing that orthogonal constraints don't systematically underestimate alignment compared to more flexible transformations.

- When MAS yields low IIA, the paper interprets this as "models don't share causal information" (e.g., Fig 3b showing different numeric tasks encode numbers differently). However, low IIA could equally result from: (a) Genuine representational differences - the models truly use different causal mechanisms, (b) Overly restrictive alignment function - shared information exists but can't be captured by orthogonal transforms, (c) Wrong layer selection - comparing non-corresponding computational stages & (d) Insufficient subspace dimensionality - d_full is set too small. The paper provides no principled way to distinguish these failure modes. This is critical because the scientific conclusions change dramatically: (a) is an interesting finding about neural computation; (b-d) are methodological limitations.

- It seemed the key changes they made that lead to MAS's better behavioral alignment judgements were 1) filtering out the behavioral null space 2) introducing an intervention-based counterfactual comparison, where the source model’s output distribution is used as the “counterfactual label.” It would be useful to show if both (1) and (2) are necessary, or if one of them is adequate, say, try an ablation of stitching + null filtering (but still comparing to the output to the target model itself rather than the source model) vs. no null space separation, but use the counterfactual loss (label given by the source model.

---

### Author Response · Authors · 2025-10-31
**Paper withdrawal**

We are withdrawing this submission because we were not able to complete satisfactory changes by the deadline. We are grateful to the reviewers and action editor for their extremely valuable feedback, which will be incorporated into future revisions. Thank you all for your time and effort!

---

### Note · Authors · 2025-10-31

**Comment:**

Thanks again!

**Withdrawal Confirmation:**

I have read and agree with the venue's withdrawal policy on behalf of myself and my co-authors.